# Impact of non-invasive oxygen reserve index versus standard SpO$_2$ monitoring on peripheral oxygen saturation during endotracheal intubation in the intensive care unit: Protocol for the randomized controlled trial NESOI2

Hugo Hille[1], Aurélie Le Thuaut[2], Pierre Asfar[3], Quentin Quelven[4], Emmanuelle Mercier[5], Anthony Le Meur[6], Jean-Pierre Quenot[7], Virginie Lemiale[8], Grégoire Muller[9,10], Martin Cour[11], Alexis Ferré[12], Asael Berge[13], Anaïs Curtiaud[14,15], Maxime Touron[16], Gaetan Plantefeve[17], Jean-Charles Chakarian[18], Jean-Damien Ricard[19], Gwenhael Colin[20], Arthur Orieux[21], Patrick Girardie[22], Mathieu Jozwiak[23,24], Manon Rouaud[2], Camille Juhel[1], Jean Reignier[25], Jean-Baptiste Lascarrou[25]*, for the CRICS-TRIGGERSEP Network[¶]

1 Medecine Intensive Reanimation, Nantes University Hospital, Nantes, France, 2 Research and Innovation Department, Methodology and Biostatistics Platform, Nantes University Hospital, Nantes, France, 3 Intensive Care Unit, Angers University Hospital, Angers, France, 4 Intensive Care Unit, Rennes University Hospital, Rennes, France, 5 Intensive Care Unit, Tours University Hospital, Tours, France, 6 Intensive Care Unit, Cholet Hospital, Cholet, France, 7 Intensive Care Unit, Dijon University Hospital, Dijon, France, 8 Intensive Care Unit, Saint-Louis University Hospital, Assistance Publique-Hôpitaux de Paris (AP-HP), Paris, France, 9 Centre Hospitalier Universitaire (CHU) d'Orléans, Médecine Intensive Réanimation, Université de Tours, MR INSERM 1327 ISCHEMIA, Université de Tours, Tours, France, 10 Clinical Research in Intensive Care and Sepsis–Trial Group for Global Evaluation and Research in Sepsis (CRICS_TRIGGERSep) French Clinical Research Infrastructure Network (F-CRIN) Research Network, Orléans, France, 11 Médecine Intensive-Réanimation, Edouard Herriot Hospital, University of Lyon, Lyon, France, 12 Intensive Care Unit, Versailles Hospital, Le Chesnay, France, 13 Intensive Care Unit, Haguenau Hospital, Haguenau, France, 14 Department of Intensive Care (Service de Médecine Intensive—Réanimation), Hôpitaux Universitaires de Strasbourg, Strasbourg, France, 15 INSERM (French National Institute of Health and Medical Research), UMR 1260, Regenerative Nanomedicine (RNM), University of Strasbourg, Strasbourg, France, 16 Intensive Care Unit, Cochin University Hospital, Assistance Publique-Hôpitaux de Paris (AP-HP), Paris, France, 17 Intensive Care Unit, Argenteuil Hospital, Argenteuil, France, 18 Service de réanimation, Centre hospitalier de Roanne, CS 80511–42328 Roanne CEDEX, Roanne, France, 19 Intensive Care Unit, Louis-Mourier Hospital, Assistance Publique-Hôpitaux de Paris (AP-HP), Colombes, France, 20 Intensive Care Unit, Vendée District Hospital, La Roche-sur-Yon, France, 21 Intensive Care Unit, Bordeaux University Hospital, Bordeaux, France, 22 Intensive Care Unit, Lille University Hospital, Lille, France, 23 Intensive Care Unit, Nice University Hospital, Nice, France, 24 UR2CA, Unité de Recherche Clinique Côte d'Azur, Université Côte d'Azur, Nice, France, 25 Nantes Université, Nantes University Hospital, Intensive Care Unit, Motion-Interactions-Performance Laboratory (MIP), UR 4334, Nantes, France

¶ The members of the CRICS-TRIGGERSEP Network are listed in the Acknowledgments section
* jeanbaptiste.lascarrou@chu-nantes.fr

**Data Availability Statement:** No datasets were generated or analysed during the current study. All

## Abstract

In critically ill patients, endotracheal intubation (ETI) is lifesaving but carries a high risk of adverse events, notably hypoxemia. Preoxygenation is performed before introducing the tube to increase the safe apnea time. Oxygenation is monitored by pulse oximeter measurement of peripheral oxygen saturation (SpO$_2$). However, SpO$_2$ is unreliable at the high

relevant data from this study will be made available upon study completion.

**Funding:** The NESOI2 trial is funded by the 2021 Programme Hospitalier de Recherche Clinique National of the French Ministry of Health (grant number PHRC-21-0202). The funder had no role in designing the trial and will have no role in collecting, analyzing, or interpreting the data; writing the trial report; or deciding to submit the trial report for publication. The funder had no role in writing the present manuscript or in deciding to submit it for publication.

**Competing interests:** JBL has received lecturing fees from BD and Masimo. Alexis Ferré reports honoraria by Fisher & Paykel for a lecture during the 2022 SFMU Congress, which bears no relation to the submitted work. None of the other authors has any competing interests to disclose. This does not alter our adherence to PLOS ONE policies on sharing data and materials.

oxygenation levels produced by preoxygenation and, in the event of desaturation, may not decrease sufficiently early to allow preventive measures. The oxygen reserve index (ORI) is a dimensionless parameter that can also be measured continuously by a fingertip monitor and reflects oxygenation in the moderate hyperoxia range. The ORI ranges from 0 to 1 when arterial oxygen saturation ($PaO_2$) varies between 100 to 200 mmHg, as occurs during preoxygenation. No trial has assessed the potential effects of ORI monitoring to guide pre-oxygenation for ETI in unstable patients. We designed a multicenter, two-arm, parallel-group, randomized, superiority, open trial in 950 critically ill adults requiring ETI. The intervention consists in monitoring ORI values and using an ORI target for preoxygenation of at least 0.6 for at least 1 minute. In the control group, preoxygenation is guided by $SpO_2$ values recorded by a standard pulse oximeter, according to the standard of care, the goal being to obtain 100% $SpO_2$ during preoxygenation, which lasts at least 3 minutes. The standard-of-care ETI technique is used in both arms. Baseline parameters, rapid-sequence induction medications, ETI devices, and physiological data are recorded. The primary outcome is the lowest $SpO_2$ value from laryngoscopy to 2 minutes after successful ETI. Secondary out-comes include cognitive function on day 28. Assuming a 10% standard deviation for the low-est $SpO_2$ value in the control group, no missing data, and crossover of 5% of patients, with the bilateral alpha risk set at 0.05, including 950 patients will provide 85% power for detect-ing a 2% between-group absolute difference in the lowest $SpO_2$ value. Should ORI monitor-ing with a target of $\geq$0.6 be found to increase the lowest $SpO_2$ value during ETI, then this trial may change current practice regarding preoxygenation for ETI.

**Trial registration**: Registered on ClinicalTrials.gov (NCT05867875) on April 27, 2023.

## Introduction

Endotracheal intubation (ETI) is performed as a lifesaving procedure in nearly a quarter of patients admitted to the intensive care unit (ICU) [1]. However, in this vulnerable population, complications occur during up to half of ETI procedures [2]. Severe complications consist of profound hypoxemia (26% of cases) and/or hypotension (25%), cardiac arrest (1%–3%), and death (0.5%–3%) [1,3,4]. Risk factors for severe hypoxemia include hypoxemia before ETI and difficult ETI [5]. Severe hypoxemia increases the risk of complications, including cardiac arrest, during ETI [3,6].

Preoxygenation is universally recommended to reduce the risk of hypoxemia by increasing the apneic time without desaturation [7,8]. However, the efficacy of preoxygenation is chal-lenging to assess. Invasive $PaO_2$ measurement is the reference standard but cannot be per-formed continuously in real time at the bedside [9]. Pulse oximetry estimation of peripheral oxygen saturation ($SpO_2$) is the oxygenation-monitoring tool most widely used during ETI in the ICU but detects hypoxemia only with a delay. Predicting hypoxemia during ETI is essential to allow timely preventive action [5]. Importantly, $SpO_2$ does not reliably reflect $PaO_2$ in the moderate hyperoxia range achieved during preoxygenation. An end-tidal oxygen fraction ($EtO_2$) greater than 90% has been suggested as a preoxygenation target [10]. However, $EtO_2$ cannot be monitored in most ICUs and may be biased by leaks around the mask [9]. More-over, in critically ill patients, notably those with acute hypoxemic respiratory failure, $EtO_2$ does not reflect oxygen reserves and therefore the effectiveness of preoxygenation [9,11]. $EtO_2$ and $SpO_2$ may be high despite low $PaO_2$ values. Finally, during acute respiratory failure, the

functional reserve capacity decreases and intrapulmonary shunting bypasses alveolar-capillary exchanges, so that $EtO_2$ does not reflect changes in $PaO_2$ [12].

Preoxygenation is usually performed for 3 to 4 minutes, but whether this duration is optimal in all patients remains unclear. In one study, increasing the duration of preoxygenation did not seem to improve efficacy [13]. An additional parameter for optimizing the efficacy of preoxygenation and predicting hypoxemia during ETI would thus be welcome.

The oxygen reserve index (ORI) is a recently introduced parameter that can be continuously monitored using the Masimo Rad-97 finger monitor, which also monitors $SpO_2$ (Masimo, Irvine, CA). This dimensionless index reflects oxygenation levels in the moderate hyperoxia range targeted by preoxygenation, defined as $PaO_2$ between 100 and 200 mmHg. The ORI can range from 0.00 ($PaO_2$ = 100 mmHg) to 1.00 ($PaO_2$ = 200 mmHg). As indicated above, $SpO_2$ does not reflect oxygenation within this range, since values $\geq$97% may indicate $PaO_2$ levels between 90 and 600 mmHg [14]. In contrast, a preoxygenation ORI target corresponding to the desired level of hyperoxia can be defined. In addition, ORI monitoring may provide an early warning that desaturation is about to occur. The potential usefulness of the ORI for this purpose has only been evaluated in small observational studies of patients undergoing scheduled surgery. ORI values dropped below 0.4 a median of 30 [20–60] seconds before the onset of desaturation [15,16]. Our recent observational NESOI study showed that, in non-hypoxemic patients undergoing ETI, ORI fell below 0.4 a median of 81 [34–146] seconds before $SpO_2$ decreased below 97% during the apnea following induction [17]. This time interval is sufficient to implement preventive measures such as mask ventilation or insertion of a supraglottic device. Interestingly, a high ORI value during preoxygenation independently predicted the absence of hypoxemia during ETI.

To assess the potential benefits of ORI monitoring during preoxygenation for ETI vs. the standard of care, we designed a multicenter randomized controlled trial in critically ill patients admitted to the ICU. The primary outcome is the lowest $SpO_2$ value recorded during ETI.

## Material and methods

This manuscript was written in accordance with Standard Protocol Items: Recommendations for Interventional Trials (SPIRIT) guidelines [18].

The first patient was recruited on August 1, 2023. On January 11, 2024, 203 patients had been included, i.e., 21.3% of the planned sample.

### Design, objectives, and setting

This protocol is for a multicenter, parallel-group, two-arm, randomized, controlled, trial comparing preoxygenation with a target ORI $\geq$0.6 for 1 minute to standard pulse-oximeter $SpO_2$ monitoring in critically ill patients undergoing ETI. The ORI is monitored using the Masimo Rad-97 monitor (Masimo, Irvine, CA, USA), which also measures $SpO_2$.

The primary objective is to determine whether ORI monitoring during ETI with the above-described target (designated "ORI monitoring" hereafter) increases the lowest $SpO_2$ value recorded by the Rad-97 monitor between the first introduction of the laryngoscope into the mouth and the end of the second minute following successful intubation. The secondary objectives are to determine whether ORI monitoring increases the lowest $SpO_2$ value that is recorded by the standard pulse oximeter (sensitivity analysis of the primary outcome) and/or that is observed in the predefined sub-groups detailed below (subgroup analysis of the primary outcome), to assess the safety of ORI monitoring based on the occurrence of immediately life-threatening adverse events as described below, and to assess the efficacy of ORI monitoring based on the efficacy outcomes listed below.

## Outcomes

The primary outcome is the lowest $SpO_2$ value measured by the Rad-97 monitor between the first introduction of the laryngoscope into the mouth and the end of the second minute following successful intubation. One of the secondary outcomes is the lowest $SpO_2$ value recorded by the standard pulse oximeter between the first introduction of the laryngoscope into the mouth and the end of the second minute following successful intubation; analysis of this outcome constitutes a sensitivity analysis of the primary outcome.

Additional secondary outcomes are the lowest $SpO_2$ values in subgroups defined based on body mass index (BMI) $<30$ vs. $\geq 30$ kg/m$^2$, hypoxemia vs. other reason for ETI, presence vs. absence of shock at inclusion, difficult vs. non-difficult ETI, and highest ORI during preoxygenation $<0.6$ vs. or $\geq 0.6$. The immediately life-threatening adverse events considered for assessing the safety of ORI monitoring are severe hypoxemia defined as $SpO_2 < 80\%$, severe hypotension defined as systolic blood pressure (SBP) $<90$ mmHg, cardiac arrest, and death. Finally, the efficacy outcomes are ICU mortality, day-28 mortality, ICU and hospital stay lengths, and cognitive status on day 28 as evaluated using the validated French version of the modified Telephone Interview for Cognitive Status (F-TICS-m) [19].

## Patients

The patients are being recruited at 20 French ICUs. Recruitment started on August 1, 2023 and is expected to end in December 2025. Consecutive adults requiring ETI in the ICU are eligible if not included in another study of ETI with an oxygen-based outcome criterion. Follow-up data are collected from trial inclusion to day 28 or death, whichever occurs first. The SPIRIT figure "Schedule of enrolment, interventions, and assessment for patients" is Fig 1.

**Inclusion criteria.** The inclusion criteria are ICU admission and a need for ETI and oxygen therapy (regardless of the device and flow rate) to obtain $SpO_2 > 97\%$.

**Non-inclusion criteria.** The non-inclusion criteria are age younger than 18 years, need for fiberoptic intubation (as determined by the attending intensivist), contraindication to laryngoscopy (e.g., unstable spinal-cord injury), insufficient time to include and randomize the patient (e.g., due to cardiac arrest), pregnancy, breastfeeding, being a correctional facility inmate, being under guardianship, not being covered by the French statutory health insurance system, and refusal of the patient (if competent) or next of kin to participate in the trial.

## Randomization

Blocked randomization in a 1:1 ratio is achieved using a computer-generated random sequence of numbers, via the secure Ennov Clinical website (https://nantes-lrsy.ennov.com/EnnovClinical). At each center, the investigators will use their identification number and password to log into the site to enroll patients. The investigators will not be aware of the blocking details. Randomization is stratified based on center, intubating intensivist experience with ETI (expert vs. non-expert as defined below), and preoxygenation device used (NIV vs. other) (Fig 2).

The Rad-97 monitor is used in both groups but the display is masked and invisible to the intubating intensivist in the control group. Blinding of the intubating physician to trial group allocation is not feasible as the Rad-97 monitor itself is visible in both groups. However, Rad-97 ORI and $SpO_2$ values are recorded for the trial by a clinical research nurse not involved in the ETI procedure or in any other components of patient management.

| TIMEPOINT** | Enrolment -t₁ | Allocation t₀ | ETI — t₁ Start of preoxygenation | ETI — t₂ Start of induction | ETI — t₃ Start of laryngoscopy | ETI — t₄ Successful ETI confirmed by capnography | ETI — t₅ End of 2nd min after successful EIT | Post-ETI follow-up t₆ Day 28 or death |
|---|---|---|---|---|---|---|---|---|
| **ENROLMENT:** | | | | | | | | |
| Eligibility screen | X | | | | | | | |
| Informed consent | X | | | | | | | |
| [List other procedures] | | | | | | | | |
| Allocation | | X | | | | | | |
| **INTERVENTIONS:** | | | | | | | | |
| [Intervention A: test intervention] | | | ORI monitoring: Target ≥0.6, 1 min. + PreO₂≥3 min SpO₂ monitoring by standard pulse oximeter and Rad-97. All three parameters available to intubator | ORI, Rad-97 SpO₂, and standard pulse oximeter SpO₂ monitoring. All three parameters available to intubator | ORI, Rad-97 SpO₂, and standard pulse oximeter SpO₂ monitoring. All three parameters available to intubator | ORI, Rad-97 SpO₂, and standard pulse oximeter SpO₂ monitoring. All three parameters available to intubator | | |
| [Intervention B: control] | | | SpO₂ monitoring by standard pulse oximetry, available to intubator. PreO₂≥3 min. ORI and Rad-97 SpO₂ recorded but not available to intubator | SpO₂ monitoring by standard pulse oximetry, available to intubator. ORI and Rad-97 SpO₂ recorded but not available to intubator | SpO₂ monitoring by standard pulse oximetry, available to intubator. ORI and Rad-97 SpO₂ recorded but not available to intubator | SpO₂ monitoring by standard pulse oximetry, available to intubator. ORI and Rad-97 SpO₂ recorded but not available to intubator | | |
| **ASSESSMENTS:** | | | | | | | | |
| [List baseline variables] | | Age Sex History Reason for ICU admission Reason for ETI BMI SCSS Diff. ETI Diff. mask vent. MACOCHA Knaus CCI | | | | | | |
| [List outcome variables] | | | | Highest ORI during Preoxygenation. **Lowest SpO₂ by Rad-97\*** Lowest standard SpO₂ Lowest SBP SpO₂<80% SBP<90 mmHg Cardiac arrest Death | → | | | ICU mortality Day 28 mortality ICU LOS Hospital LOS F-TICS-m on day-28 survivors |
| [List other data variables] | | | SBP DBP MAP HR Vasopressors Standard SpO₂ ORI Rad-97 SpO₂ ORI RR EtCO₂ FiO₂ PEP VT Preoxygenation device Duration of preoxygenation | SBP DBP MAP HR Vasopressors Standard SpO₂ ORI Rad-97 SpO₂ ORI RR EtCO₂ FiO₂ PEP VT Oxygenation device if any Drugs and doses | SBP DBP MAP HR Vasopressors Standard SpO₂ ORI Rad-97 SpO₂ ORI RR EtCO₂ FiO₂ PEP VT Laryngoscopy device Intubation device Sellick maneuver, yes/no | SBP DBP MAP HR Vasopressors Standard SpO₂ ORI Rad-97 SpO₂ ORI RR EtCO₂ FiO₂ PEP VT | SBP DBP MAP HR Vasopressors Standard SpO₂ ORI Rad-97 SpO₂ ORI RR EtCO₂ FiO₂ PEP VT | |
| | | | | Reventilation, yes/no Cormack-Lehane grade POGO VIDIAC N of attempts | | | | |
| | | | | Adverse events | → | | | |

ICU: intensive care unit; ETI: endotracheal intubation; BMI: body mass index; SCSS: skin color scale score; Diff. ETI: criteria predicting difficult ETI; Diff. mask vent.: criteria predicting difficult mask ventilation; CCI: Charlson's Comorbidity Index; SBP: systolic blood pressure; DBP: diastolic blood pressure; MAP: mean arterial pressure; HR: heart rate; SpO₂: peripheral oxygen saturation; ORI: oxygen reserve index; RR: respiratory rate; EtCO₂: expiratory end-tidal carbon dioxide concentration; FiO₂: fraction of inspired oxygen; PEP: positive end-expiratory pressure; VT: tidal volume; F-TICS-m: modified Telephone Interview for Cognitive Status; Rad-97: Masimo Rad-97 monitor for concomitant ORI and SpO₂ measurement; LOS: length of stay; POGO: percentage of glottic opening; VIDIAC: videolaryngoscopic intubation and difficult airway classification

*primary outcome

**Fig 1. SPIRIT schedule of enrolment, interventions, and assessments.**

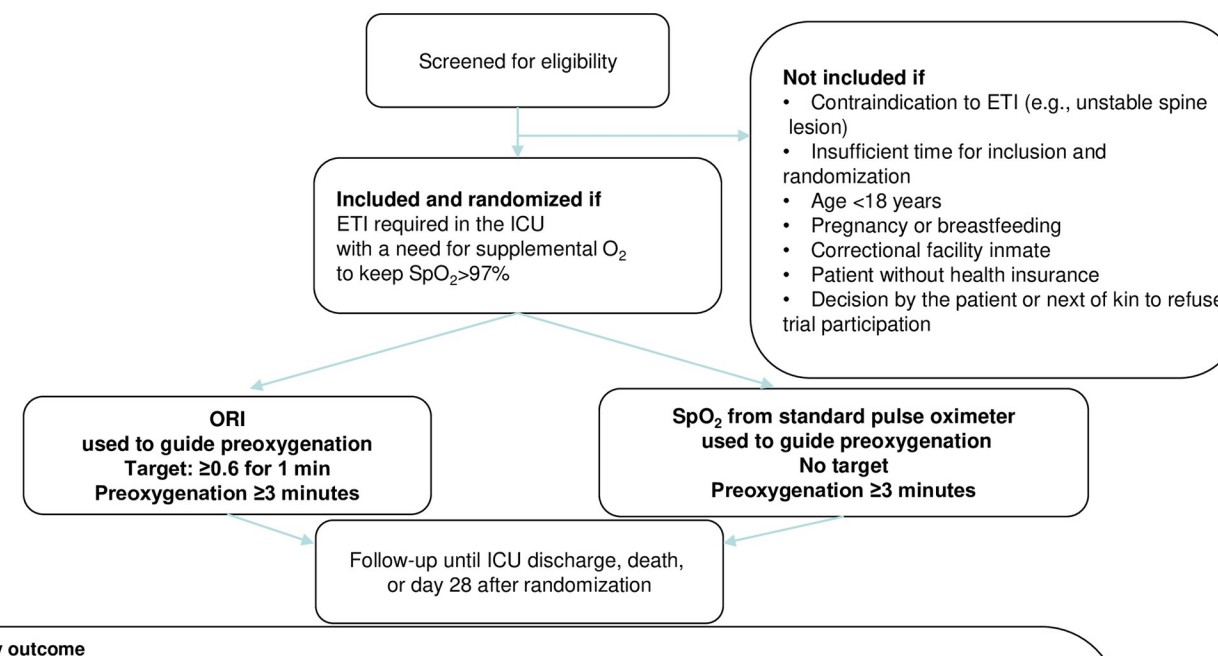

**Fig 2. Patient flowchart.**

## Intervention and control arms

**Intervention arm.** The trial intervention consists in monitoring the ORI to achieve a value ≥0.6 for 1 minute during preoxygenation. The end of preoxygenation is defined as the start of induction, which occurs when the preoxygenation duration is at least 3 minutes and the ORI has been ≥0.6 for at least 1 minute. Thus, if ORI is ≥0.6 after 90 seconds of preoxygenation, induction is started 90 seconds later, after 3 minutes of preoxygenation. If ORI is ≥0.6 after 2.5 minutes of preoxygenation, induction is started 1 minute later, after 3.5 minutes of preoxygenation. In patients whose ORI remains <0.6 after 3 minutes of preoxygenation, the preoxygenation device is changed and preoxygenation continued until ORI is ≥0.6 for at least 1 minute. When this goal is not achieved, no maximal preoxygenation duration is specified given the pragmatic trial design; however, the investigators have been informed than 6 to 8 minutes of preoxygenation are expected to maximize benefits) (Fig 3). The ORI threshold of 0.6, corresponding to an estimated $PaO_2$ value of 160 mmHg, was chosen based on

# Intervention arm

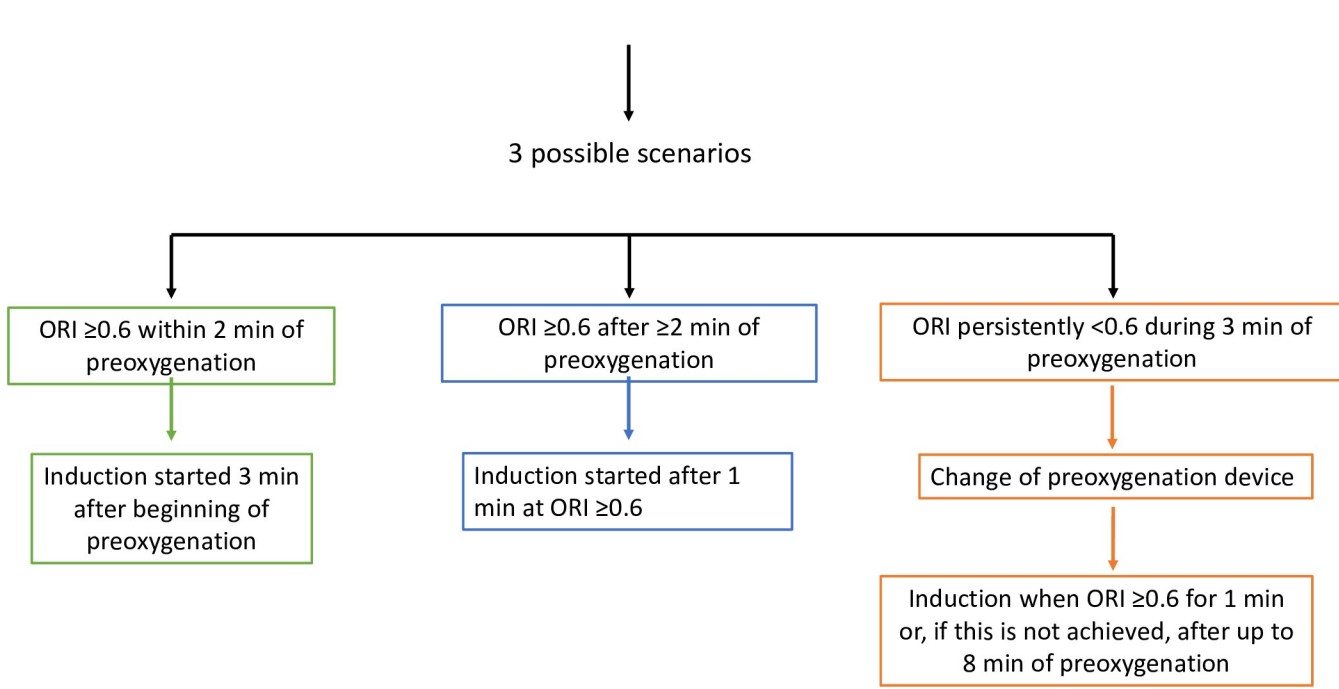

**Fig 3. Details of the intervention.**

unpublished personal data (JBL). The SpO$_2$ values recorded by the standard pulse oximeter are also communicated to the intubating intensivist.

**Control arm.**   In the control group, oxygenation is monitored according to the standard of care, that is, using the standard pulse oximeter available in each ICU (Fig 4A and 4B). The intubating intensivist will not be aware of the SpO$_2$ and ORI values provided by the Rad-97 monitor throughout the ETI procedure, which are recorded by the clinical research nurse. The Rad-97 monitor is masked and its alarms turned off. Preoxygenation lasts at least 3 minutes, with no maximum duration.

**(A)** Control arm. **(B)** Intervention arm.

**Standard of care applied in both arms.**   The other components of the standard of care are applied in both arms. Two intensivists are present throughout the entire ETI procedure, including at least one ETI expert defined as having at least 5 years of ICU experience or at least 1 year of ICU experience plus at least 2 years of operating-room experience [20]. At least one of the two intensivists is trained in ORI monitoring. One of the two intensivists is the physician in charge of the patient. Preoxygenation is not started until the standard pulse oximeter and Rad-97 monitor are in place, on different fingers (ideally the forefinger for the standard pulse oximeter and the middle finger for the Rad-97 monitor) of the hand on the side contralateral to the arm equipped with the non-invasive blood-pressure cuff.

Selection of the preoxygenation device is at the discretion of the intubating intensivist. The options are a bag-valve mask with a flow rate of 60 L/min [21–23], non-rebreather mask with a flow rate of 60 L/min [23,24], NIV with 100% fraction of inspired oxygen (FiO$_2$) [25], and high-flow nasal cannula oxygen with 100% FiO$_2$ and a flow rate of 60 L/min [26,27]. However, the protocol recommends NIV in patients with hypoxemia defined as a need for >8 L/min of supplemental oxygen to maintain SpO$_2$ ≥95%. Additional apneic oxygenation or/and apneic ventilation are allowed [28].

# Control arm

Standard pulse oximeter used locally
SpO$_2$ values recorded by a clinical research nurse and
communicated to the intubator

Preoxygenation for at least 3
minutes

Induction          Intubation

Rad-97 monitor
ORI and SpO$_2$ values recorded by the same clinical research
nurse but not communicated to the intubator

# Intervention Arm

Standard pulse oximeter used locally
SpO2 values recorded by a clinical research nurse and
communicated to the intubator

Preoxygenation for at least 3
minutes
(No upper limit)

Induction started IF preoxygenation ≥3 min
AND ORI≥0.6 for 1 min, then intubation

Rad-97 monitor
ORI and SpO2 values recorded by the same clinical
research nurse and communicated to the intubator

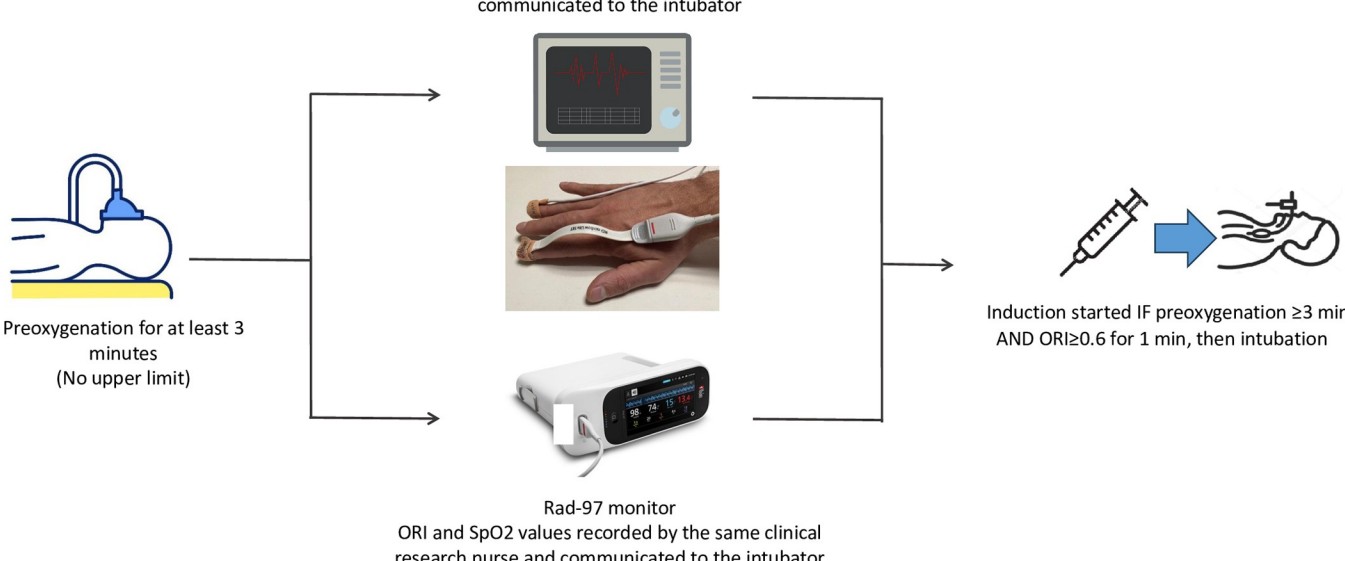

**Fig 4. Details of endotracheal intubation management in the control and intervention arms**

The choice of induction drugs and their dosages is at the discretion of the intubating intensivist. However, the intubating intensivists are encouraged to follow international [7] and French [8] recommendations that etomidate (0.2–0.3 mg/kg) and ketamine (1–2 mg/kg) be

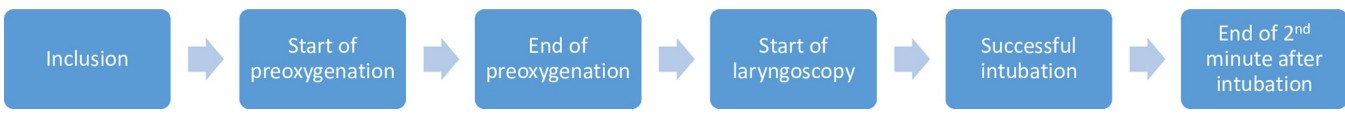

**Fig 5. Data collection timepoints.**

used as hypnotics, succinylcholine (1 mg/kg) as the first-line muscle relaxant, and rocuronium (1 mg/kg) as the alternative muscle relaxant.

Selection of the intubation device (laryngoscope or videolaryngoscope) and of blade type and size is at the discretion of the intubating intensivist. If the first attempt fails, the intubating intensivist is free to choose between repeating the same technique or switching to a different technique. The choice of this alternative technique is at the discretion of the intubating intensivist, who follows recommendations [7,8]. Each insertion of the laryngoscope into the mouth is counted as an intubation attempt. Correct tube position is confirmed based on examination of four consecutive capnography cycles. Immediately after successful ETI, the balloon is inflated and the tube connected to the ventilator.

## Data collection

At each participating ICU and for each patient, the trial data are entered into an electronic case report form (Ennov, Paris, France), in real time, by a clinical nurse not involved in patient management. The collected baseline data include demographics, medical history, reason for ICU admission, reason for ETI, BMI, skin color scale score (Fitzpatrick score [29]), criteria predicting difficult ETI and difficult mask ventilation [30], MACOCHA score [30], Knaus chronic disease score [31], and Charlson Comorbidity Index [32]. The Sequential Organ Failure Assessment (SOFA) score [33] and Simplified Acute Physiology Score version II (SAPS II) [34] are recorded at ICU admission. The following are collected at inclusion and throughout the ETI procedure: hemodynamic data (SBP, diastolic blood pressure [DBP], mean arterial pressure [MAP], heart rate, and need for vasopressor therapy) and respiratory data (SpO$_2$, ORI, respiratory rate, expiratory EtCO$_2$, FiO$_2$, positive end-expiratory pressure, and tidal volume). The data collection timepoints during ETI are inclusion, start of preoxygenation, end of preoxygenation (which coincides with start of induction, i.e., of hypnotic administration), start of laryngoscopy defined as the first introduction of the laryngoscope into the mouth, confirmation of successful intubation by capnography, and end of the second minute after successful intubation. ETI duration runs from induction initiation to confirmation of successful intubation (Fig 5). SpO$_2$ values are recorded from both the standard pulse oximeter and the Rad-97 monitor. The lowest SpO$_2$ values recorded by each device during the ETI procedure, the highest ORI value during preoxygenation, and the lowest SBP value during the ETI procedure are collected. Other routinely collected data are the preoxygenation device used; oxygenation device used between induction initiation and beginning of laryngoscopy, if any; duration of preoxygenation; drugs and doses used; intubation device used; need to perform a Sellick maneuver and/or re-ventilation (via a face mask or supraglottic device); Cormack-Lehane grade [35], POGO [36], and VIDIAC scores [37]; number of attempts before successful intubation (each attempt being defined as laryngoscope insertion into the mouth), with details; and adverse events during ETI.

Each patient is followed up until death or day 28, whichever occurs first. The following are recorded: SOFA score every day from day 1 to day 7, vital status at ICU discharge and on day 28, ICU and hospital stay lengths, and cognitive status on day 28 assessed during a telephone interview using the F-TICS-m [19].

## Interim analysis

No interim analysis is planned.

## Ethics and dissemination

**Ethics.** The NESOI-2 protocol was approved by the appropriate ethics committee (*Comité de Protection des Personnes Ouest III*) on March 20, 2023 (#2023-A00114-41) and was registered on ClinicalTrials.gov on April 27, 2023 (#NCT05867875). Before trial inclusion, the patient, or next of kin if the patient is incompetent, is given an information document. Written or oral consent is then sought. If the patient is incompetent and no next of kin is available, trial inclusion is performed according to the emergency procedure set forth by French law, and informed consent is then sought as soon as a relative is available then when the patient recovers competency.

**Dissemination.** The policy for publishing the trial findings will comply with international recommendations [38] and the CONSORT statement (http://www.consort-statement.org). The findings will be published in peer-reviewed journals and presented during national and international scientific meetings. Communications and scientific reports relevant to the trial will be under the responsibility of the study coordinator (JBL), who will first obtain the approval of the other investigators. Guidelines for authorship issued by the International Committee of Medical Journal Editors will be followed.

## Patient and public involvement

Neither the patients nor the public were involved in designing this protocol. They will not be involved in disseminating the findings.

## Statistics

**Sample size.** Assuming a standard deviation of 10% for the lowest $SpO_2$ value in the control group [1], no missing data, and crossover of 5% of patients, with the bilateral alpha risk set at 0.05, the inclusion of 950 patients provides 85% power for detecting a 2% absolute between-group difference in the lowest $SpO_2$ value.

The recruitment rate predicted based on data from each participating center was 5 patients per month. The observed recruitment rate from trial initiation on August 1, 2023, to January 11, 2024, is consistent with this prediction.

**Statistical analyses.** The baseline characteristics of the overall population and each randomization group will be described as absolute number (%) for categorical variables and as mean±SD, range, and interquartile range for quantitative variables. No statistical tests will be performed to compare the two groups at baseline.

The main analysis will be based on the modified intention-to-treat population defined as all randomized patients who meet the legal criteria. Patients with $SpO_2$ values below 98% at the end of preoxygenation will be included in the modified intention-to-treat population. An analysis will also be performed in the per-protocol population obtained by excluding patients with ORI recording failure or ORI<0.6 at the end of preoxygenation.

The primary outcome will be compared between groups using a mixed-effects linear regression model to account for the stratification variables (center as a random effect and operator experience and preoxygenation device as fixed effects). In the event of non-normality of the residuals observed graphically (Q-Q plot), the usual transformations of the variables will be tested to improve the fit.The analyses of the effect of ORI monitoring on the primary outcome will be repeated in sub-groups defined by BMI (< or ≥30 kg/m$^2$), reason for ETI (hypoxemia

vs. other), presence vs. absence of shock at inclusion, difficult vs. non-difficult ETI, and ORI<0.6 vs. ≥0.6. For each subgroup analysis, the statistical model described for the main analysis will be used, with incorporation of the interaction terms (subgroup variable by randomization group). The same statistical model will also be used to compare the groups regarding the lowest $SpO_2$ value recorded by the standard pulse oximeter.

The proportions of patients with at least one serious adverse event threatening short-term survival will be compared using a mixed-effects linear regression model. ICU and day-28 mortality will be evaluated by plotting Kaplan-Meier curves and compared between groups using a Cox proportional hazards model. Goodness of fit of the model will be assessed using Schoenfeld residuals to study the proportional hazard assumption and Martingale residuals to study linearity of continuous covariates. A Fine-and-Gray model will be used to compare ICU and hospital stay lengths. Cognitive status on day 28 will be compared between groups using a mixed-effects linear regression model in the population of day-28 survivors. All analyses will be adjusted on the stratification criteria.

For all analyses, *P* values ≤0.05 will be taken to indicate significant differences. The statistical analyses will be performed using SAS software v9.4 (SAS Institute, Cary, NC).

In case of missing data for the primary outcome, multiple imputation will be performed. No imputation will be performed for missing data about secondary outcomes.

**Harms.** Adverse events are recorded into the electronic case report form for each patient. Patients who experience one or more adverse events possibly related to ETI may be temporarily withdrawn from the trial, if deemed appropriate by the attending intensivist.

## Discussion

Several recent trials have assessed the efficacy of various interventions for improving the safety of ETI in highly unstable patients [4,28,39–41]. Promising results have been obtained [17]. Videolaryngoscopy is gaining in popularity [42,43]. The intervention in our trial consists in ORI monitoring as a means of limiting desaturation during ETI. The trial thus assesses a single component of the ETI procedure, which involves many more. Whether focusing on a single component will allow the detection of a significant effect is the main issue raised by our protocol.

$SpO_2$ monitoring is widely used in the ICU despite the absence of randomized controlled trials demonstrating that this practice decreases mortality. We believe it would be unacceptable to abstain from using $SpO_2$ monitoring. Thus, all patients will have their $SpO_2$ values monitored using a standard pulse oximeter throughout the ETI procedure, and the results will be communicated to the intubating intensivists in both trial arms. Mean $SpO_2$ values obtained using standard pulse oximeters correlated well with those obtained using the Rad-97 monitor. It should be noted that relying solely on $SpO_2$ values for monitoring oxygenation remains controversial [44]. The primary outcome (lowest $SpO_2$) is recorded from the beginning of laryngoscopy to the end of the second minute after successful intubation. This duration was chosen to avoid interactions with ventilator setting changes, in accordance with a previous study [28]. Preoxygenation devices are important to the safety of intubation [45]. Randomization will occur before preoxygenation. Therefore, for patients in the ORI-monitoring group whose ORI does not reach 0.6, the protocol allows a change to another preoxygenation device.

We chose an ORI cutoff of 0.6 as the preoxygenation target for several reasons. First, in an intraoperative study, 96.6% of $PaO_2$ values were ≥150 mmHg when ORI was over 0.55 [46]. Second, in the NESOI1 study, 0.6 was the median ORI value at the end of preoxygenation (0.62 [0.26–0.83]) [17]. In the NESOI2 study, the median ORI in patients without an $SpO_2$ drop below 90% during intubation was close to 0.6 (0.68 [0.27–0.89]). Moreover, 0.6 is midway

between values in patients without vs. with $SpO_2<97\%$ in NESOI1 (0.77 [0.59–1.00] vs. 0.49 [0.22–0.77]). Thus, an ORI of 0.6 indicates an effective level of hyperoxia while also probably being achievable in a large-scale clinical study.

## Contact for public queries

General queries including information about current recruitment status can be addressed to Ms. Florance Pasquier, florane.pasquier@chu-nantes.fr

## Contact for scientific queries

The primary investigator is Jean-Baptiste Lascarrou, who will deal with any scientific enquiries.

Dr. Jean-Baptiste Lascarrou, Médecine Intensive Réanimation, CHU de Nantes, 30 Bd Jean Monet, 44093 Nantes Cedex 9, FRANCE

Telephone: +33 240 087 376
jeanbaptiste.lascarrou@chu-nantes.fr

## Public title

New non-invasive oxygenation parameter compared to standard non-invasive pulse oximetry for preventing low oxygenation during endotracheal intubation

## Supporting information

**S1 Checklist. SPIRIT 2013 checklist: Recommended items to address in a clinical trial protocol and related documents\*.**
(DOC)

**S1 File.**
(DOCX)

## Acknowledgments

We are indebted to Antoinette Wolfe, MD, for assistance in preparing and reviewing the manuscript; Joseph Herault for managing the database; Manon Rouaud for coordinating the trial; and Emmanuel Billaud and Alban Jauffrit, both at Masimo Corporation, for providing details on how the Rad-97 monitor works.

Members of the CRICS-TRIGGERSEP Network:

Hugo Hille, Médecine Intensive Reanimation, Nantes University Hospital, Nantes, France
Pierre Asfar, Intensive Care Unit, Angers University Hospital, Angers, France
Emmanuelle Mercier, Intensive Care Unit, Tours University Hospital, Tours, France
Jean-Pierre Quenot, Intensive Care Unit, Dijon University Hospital, Dijon, France
Grégoire Muller, Médecine Intensive Réanimation, Orléans University Hospital, MR INSERM 1327 ISCHEMIA, Université de Tours, Tours, France

Asael Berge, Intensive Care Unit, Haguenau Hospital, Haguenau, France
Anaïs Curtiaud, Médecine Intensive Réanimation), Strasbourg University Hospital, Strasbourg, France; INSERM (French National Institute of Health and Medical Research), UMR 1260, Regenerative Nanomedicine (RNM), Strasbourg University, Strasbourg, France

Maxime Touron, Intensive Care Unit, Cochin University Hospital, Assistance Publique-Hôpitaux de Paris (AP-HP), Paris, France

Gaetan Plantefeve, Intensive Care Unit, Argenteuil Hospital, Argenteuil, France

Gwenhael Colin, Intensive Care Unit, Vendée District Hospital, La Roche-sur-Yon, France

Jean Reignier, Intensive Care Unit, Nantes University Hospital, Motion-Interactions-Performance Laboratory (MIP), UR 4334, Nantes, France

Jean-Baptiste Lascarrou, Intensive Care Unit, Nantes University Hospital, Motion-Interactions-Performance Laboratory (MIP), UR 4334, Nantes, France (contact for the Network)

## Author Contributions

**Conceptualization:** Hugo Hille, Manon Rouaud, Jean Reignier, Jean-Baptiste Lascarrou.

**Data curation:** Aurélie Le Thuaut, Jean-Baptiste Lascarrou.

**Formal analysis:** Aurélie Le Thuaut, Jean-Baptiste Lascarrou.

**Funding acquisition:** Jean-Baptiste Lascarrou.

**Investigation:** Hugo Hille, Pierre Asfar, Quentin Quelven, Emmanuelle Mercier, Anthony Le Meur, Jean-Pierre Quenot, Virginie Lemiale, Grégoire Muller, Martin Cour, Alexis Ferré, Asael Berge, Anaïs Curtiaud, Maxime Touron, Gaetan Plantefeve, Jean-Charles Chakarian, Jean-Damien Ricard, Gwenhael Colin, Arthur Orieux, Patrick Girardie, Mathieu Jozwiak, Camille Juhel, Jean Reignier, Jean-Baptiste Lascarrou.

**Methodology:** Hugo Hille, Aurélie Le Thuaut, Manon Rouaud, Camille Juhel, Jean-Baptiste Lascarrou.

**Project administration:** Manon Rouaud, Camille Juhel, Jean Reignier, Jean-Baptiste Lascarrou.

**Resources:** Jean-Baptiste Lascarrou.

**Software:** Jean-Baptiste Lascarrou.

**Supervision:** Manon Rouaud, Camille Juhel, Jean-Baptiste Lascarrou.

**Validation:** Manon Rouaud, Jean-Baptiste Lascarrou.

**Visualization:** Camille Juhel, Jean-Baptiste Lascarrou.

**Writing – original draft:** Hugo Hille, Jean-Baptiste Lascarrou.

**Writing – review & editing:** Pierre Asfar, Quentin Quelven, Emmanuelle Mercier, Anthony Le Meur, Jean-Pierre Quenot, Virginie Lemiale, Grégoire Muller, Martin Cour, Alexis Ferré, Asael Berge, Anaïs Curtiaud, Maxime Touron, Gaetan Plantefeve, Jean-Charles Chakarian, Jean-Damien Ricard, Gwenhael Colin, Arthur Orieux, Patrick Girardie, Mathieu Jozwiak, Manon Rouaud, Camille Juhel, Jean Reignier, Jean-Baptiste Lascarrou.

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
