## [Decision Letter · Decision Letter 0]

6 Mar 2024

PONE-D-24-03563Impact of non-invasive oxygen reserve index versus standard SpO2 monitoring on peripheral oxygen saturation during endotracheal intubation in the intensive care unit: Protocol for the randomized controlled trial NESOI2

PLOS ONE

Dear Dr. Lascarrou,

Thank you for submitting your manuscript to PLOS ONE. After careful consideration, we feel that it has merit but does not fully meet PLOS ONE’s publication criteria as it currently stands. Therefore, we invite you to submit a revised version of the manuscript that addresses the points raised during the review process.

We look forward to receiving your revised manuscript.

Kind regards,

Ryo Yamamoto

Academic Editor

PLOS ONE

“French Ministry of Health.

PHRC 2021.”

“JBL has received lecturing fees from BD and Masimo. Alexis Ferré reports honoraria by Fisher & Paykel for a lecture during SFMU Congress 2022, outside the submitted work. All other authors reports not competing interests.”

5. One of the noted authors is a group or consortium [CRICS-TRIGGERSEP Network]. In addition to naming the author group, please list the individual authors and affiliations within this group in the acknowledgments section of your manuscript. Please also indicate clearly a lead author for this group along with a contact email address.

6. We note that Figure 4 in your submission contain copyrighted images. All PLOS content is published under the Creative Commons Attribution License (CC BY 4.0), which means that the manuscript, images, and Supporting Information files will be freely available online, and any third party is permitted to access, download, copy, distribute, and use these materials in any way, even commercially, with proper attribution. For more information, see our copyright guidelines: http://journals.plos.org/plosone/s/licenses-and-copyright.

1. You may seek permission from the original copyright holder of Figure 4 to publish the content specifically under the CC BY 4.0 license.

7. We note that the original protocol that you have uploaded as a Supporting Information file contains an institutional logo. As this logo is likely copyrighted, we ask that you please remove it from this file and upload an updated version upon resubmission.

Additional Editor Comments:

Thank you for submitting the protocol of interesting study.

Please see reviewer's comment.

Reviewers' comments:

Reviewer's Responses to Questions

**Comments to the Author**

1. Does the manuscript provide a valid rationale for the proposed study, with clearly identified and justified research questions?

Reviewer #1: Yes

Reviewer #2: Yes

2. Is the protocol technically sound and planned in a manner that will lead to a meaningful outcome and allow testing the stated hypotheses?

Reviewer #1: Partly

Reviewer #2: Yes

3. Is the methodology feasible and described in sufficient detail to allow the work to be replicable?

Reviewer #1: Yes

Reviewer #2: Yes

4. Have the authors described where all data underlying the findings will be made available when the study is complete?

Reviewer #1: Yes

Reviewer #2: Yes

5. Is the manuscript presented in an intelligible fashion and written in standard English?

Reviewer #1: Yes

Reviewer #2: Yes

6. Review Comments to the Author

You may also provide optional suggestions and comments to authors that they might find helpful in planning their study.

Reviewer #1: Thank you for sharing the manuscript titled "Impact of Non-Invasive Oxygen Reserve Index Versus Standard SpO2 Monitoring on Peripheral Oxygen Saturation During Endotracheal Intubation in the Intensive Care Unit: Protocol for the Randomized Controlled Trial NESOI2" for review.

The paper outlines a research protocol assessing the impact of preoxygenation guided by the Oxygen Reserve Index (ORI) on the severity of desaturation during orotracheal intubation in ICU settings.

The manuscript is well-written and largely unambiguous. The methodology is thoroughly described and detailed. I have no significant comments.

Abstract

Line 85: "ORI target for preoxygenation of at least 0.6 for at least 1 minute." Please explain the rationale for selecting this specific target.

Line 85: "Preoxygenation is guided by SpO2 values recorded by a standard pulse oximeter, according to the standard of care." Could you describe how SpO2 guides preoxygenation? (Is it aiming for a SpO2 of 100% for a specific duration?).

Main Manuscript

• Line 189: Please explain the relevance of reference 19 in the methodology. Does it describe a study with a similar methodology? Or is it related to another aspect? In any case, if the reference is cited to justify methodological choices (such as the rationale behind the decision to monitor saturation 2 minutes post-intubation), it would be more appropriate to include it in the discussion section.

• Line 190: "The secondary outcome includes the sensitivity analysis of the primary outcome." This analysis pertains to alternative statistical presentations of the primary endpoint. I suggest relocating this alongside the calculations for the primary endpoint.

• Line 257: "Chosen based on unpublished personal data (JBL)." Please provide a brief description of what these data have shown.

• Line 264: "The intensivist will not be aware of the SpO2 and ORI values..." If the intensivist is unaware of these monitors, on what basis will he determine the duration of preoxygenation?

• Line 291: "Intubating intensivist..." If the choice of drugs is left to the discretion of the intensivist, I see no value in citing recommendations.

• Line 322: The various phases, durations, and definitions could be graphically represented on a timeline, enhancing understanding.

• The choice of alternative devices for initial preoxygenation should be controlled. Suppose the distribution of alternatives to preoxygenation differs between groups. In that case, it might raise questions about whether the difference in preoxygenation quality is related to the monitoring or the preoxygenation tool.

Reviewer #2: Impact of non-invasive oxygen reserve index versus standard SpO2 monitoring on peripheral oxygen saturation during endotracheal intubation in the intensive care unit: Protocol for the randomized controlled trial NESOI

Thank you for considering me for this review.

This is a protocol for a multicenter, two-arm, 83 parallel-group, randomized, superiority, open trial in 950 critically ill adults requiring endo tracheal intubation (ETI) aiming at monitoring ORI values and using an oxygen reserve index (ORI) target for 85 preoxygenation of at least 0.6 for at least 1 minute as compared to a standard of care SpO2 monitoring.

The protocol is well written and could have significant implications on the care of this vulnerable patient group by potentially preventing desaturations.

Here are my comments:

• ICU patients needing intubation are a heterogenous population. I think including a respiratory failure patient or a congestive cardiac patient with a trauma or septic patient would skew the results. The study would benefit from a more homogenous study population.

• Since ORI values represent a PaO2 of >100mmHg none of the patients with SpO2 <98%, will have a measurable ORI value. Therefore, I would suggest explicitly excluding patients who do not reach an SpO2 of 99-100% after 3-5 min of preoxygenation.

• P 9, first paragraph – the primary outcome measurement time (lowest SpO2 value, first introduction of the laryngoscope into the mouth and the end of the second 189 minute following successful intubation) seems too short to me. Two minutes after intubation could be too short for SpO2 to recover after apnea in critically ill patients.

• P 9, R 197 – As mentioned above, ORI will be 0 at SpO2 >98%, thus it can’t assess severe hypoxic events

7. PLOS authors have the option to publish the peer review history of their article (what does this mean?). If published, this will include your full peer review and any attached files.

Reviewer #1: **Yes: **Issam Tanoubi

Reviewer #2: No

---

## [Author Response · Author response to Decision Letter 0]

25 Apr 2024

Please find enclosed Reviewer Answer.

---

## [Decision Letter · Decision Letter 1]

29 May 2024

PONE-D-24-03563R1Impact of non-invasive oxygen reserve index versus standard SpO2 monitoring on peripheral oxygen saturation during endotracheal intubation in the intensive care unit: Protocol for the randomized controlled trial NESOI2PLOS ONE

Dear Dr. Lascarrou,

Thank you for submitting your manuscript to PLOS ONE. After careful consideration, we feel that it has merit but does not fully meet PLOS ONE’s publication criteria as it currently stands. Therefore, we invite you to submit a revised version of the manuscript that addresses the points raised during the review process.

We look forward to receiving your revised manuscript.

Kind regards,

Ryo Yamamoto

Academic Editor

PLOS ONE

Journal Requirements:

Reviewers' comments:

Reviewer's Responses to Questions

**Comments to the Author**

1. Does the manuscript provide a valid rationale for the proposed study, with clearly identified and justified research questions?

Reviewer #1: Yes

Reviewer #2: Yes

2. Is the protocol technically sound and planned in a manner that will lead to a meaningful outcome and allow testing the stated hypotheses?

Reviewer #1: Yes

Reviewer #2: Yes

3. Is the methodology feasible and described in sufficient detail to allow the work to be replicable?

Reviewer #1: Yes

Reviewer #2: Yes

4. Have the authors described where all data underlying the findings will be made available when the study is complete?

Reviewer #1: Yes

Reviewer #2: Yes

5. Is the manuscript presented in an intelligible fashion and written in standard English?

Reviewer #1: Yes

Reviewer #2: Yes

6. Review Comments to the Author

You may also provide optional suggestions and comments to authors that they might find helpful in planning their study.

Reviewer #1: Thank you for your revisions and for addressing the initial comments on your manuscript. Upon further review, I have two additional suggestions that could strengthen your submission:

Responses to comments 1, 5, and 9 from Reviewer #1 should be incorporated into the discussion section of the article to better justify and explain the methodological choices made.

Concerning Reviewer #2’s Comments: I understand that the authors prefer not to exclude patients with SpO₂ below 98% at the end of preoxygenation due to their rarity. However, it is essential to clearly describe in the statistical section how their data will be analyzed should this issue arise.

Reviewer #2: The responses to my initial review have been answered. I accept the author's comments and I agree with the publication

7. PLOS authors have the option to publish the peer review history of their article (what does this mean?). If published, this will include your full peer review and any attached files.

Reviewer #1: **Yes: **Issam Tanoubi

Reviewer #2: **Yes: **Peter Szmuk, M.D.

---

## [Author Response · Author response to Decision Letter 1]

4 Jun 2024

Please find enclosed Reviewer Answers.

---

## [Decision Letter · Decision Letter 2]

2 Jul 2024

PONE-D-24-03563R2Impact of non-invasive oxygen reserve index versus standard SpO2 monitoring on peripheral oxygen saturation during endotracheal intubation in the intensive care unit: Protocol for the randomized controlled trial NESOI2PLOS ONE

Dear Dr. Lascarrou,

Thank you for submitting your manuscript to PLOS ONE. After careful consideration, we feel that it has merit but does not fully meet PLOS ONE’s publication criteria as it currently stands. Therefore, we invite you to submit a revised version of the manuscript that addresses the points raised during the review process.

An additional statistical reviewer suggest some minor revisions.

Please respond to them.

We look forward to receiving your revised manuscript.

Kind regards,

Ryo Yamamoto

Academic Editor

PLOS ONE

Journal Requirements:

Reviewers' comments:

Reviewer's Responses to Questions

**Comments to the Author**

1. Does the manuscript provide a valid rationale for the proposed study, with clearly identified and justified research questions?

Reviewer #1: Yes

Reviewer #3: Yes

2. Is the protocol technically sound and planned in a manner that will lead to a meaningful outcome and allow testing the stated hypotheses?

Reviewer #1: Yes

Reviewer #3: Yes

3. Is the methodology feasible and described in sufficient detail to allow the work to be replicable?

Reviewer #1: Yes

Reviewer #3: Yes

4. Have the authors described where all data underlying the findings will be made available when the study is complete?

Reviewer #1: Yes

Reviewer #3: Yes

5. Is the manuscript presented in an intelligible fashion and written in standard English?

Reviewer #1: Yes

Reviewer #3: Yes

6. Review Comments to the Author

You may also provide optional suggestions and comments to authors that they might find helpful in planning their study.

Reviewer #1: I thank the authors for adequately addressing the reviewers' comments. I have no further comments to add.

Reviewer #3: On the overall, the manuscript addresses previously raised questions. My additional comments are as follows:

1. A linear mixed model will be fitted, which is based on Gaussian assumptions of the random terms (random effects & errors). What would be the recourse, when the Gaussian assumptions fail (which can certainly happen)?

2. Cox regression will be employed, but necessary goodness of fit statistics and tests of the proportional hazards assumptions need to be mentioned.

7. PLOS authors have the option to publish the peer review history of their article (what does this mean?). If published, this will include your full peer review and any attached files.

Reviewer #1: **Yes: **Issam Tanoubi

Reviewer #3: No

---

## [Decision Letter · Decision Letter 3]

10 Jul 2024

Impact of non-invasive oxygen reserve index versus standard SpO2 monitoring on peripheral oxygen saturation during endotracheal intubation in the intensive care unit: Protocol for the randomized controlled trial NESOI2

PONE-D-24-03563R3

Dear Dr. Lascarrou,

We’re pleased to inform you that your manuscript has been judged scientifically suitable for publication and will be formally accepted for publication once it meets all outstanding technical requirements.

Kind regards,

Ryo Yamamoto

Academic Editor

PLOS ONE

Additional Editor Comments (optional):

Reviewers' comments:

Reviewer's Responses to Questions

**Comments to the Author**

1. Does the manuscript provide a valid rationale for the proposed study, with clearly identified and justified research questions?

Reviewer #3: Yes

2. Is the protocol technically sound and planned in a manner that will lead to a meaningful outcome and allow testing the stated hypotheses?

Reviewer #3: Yes

3. Is the methodology feasible and described in sufficient detail to allow the work to be replicable?

Reviewer #3: Yes

4. Have the authors described where all data underlying the findings will be made available when the study is complete?

Reviewer #3: Yes

5. Is the manuscript presented in an intelligible fashion and written in standard English?

Reviewer #3: Yes

6. Review Comments to the Author

You may also provide optional suggestions and comments to authors that they might find helpful in planning their study.

Reviewer #3: The authors were able to address my previous round of questions. I have no further questions this round.

7. PLOS authors have the option to publish the peer review history of their article (what does this mean?). If published, this will include your full peer review and any attached files.

Reviewer #3: No

---

## [Editor Report · Acceptance letter]

1 Aug 2024

PONE-D-24-03563R3 

PLOS ONE

Dear Dr. Lascarrou, 

I'm pleased to inform you that your manuscript has been deemed suitable for publication in PLOS ONE. Congratulations! Your manuscript is now being handed over to our production team.

Kind regards, 

on behalf of

Dr. Ryo Yamamoto 

Academic Editor

PLOS ONE